# Thermodynamics of the Ramsey Zone

**DOI:** 10.3390/e25101430

**Published:** 2023-10-10

**Authors:** Rogério Jorge de Assis, Ciro Micheletti Diniz, Norton Gomes de Almeida, Celso Jorge Villas-Bôas

**Affiliations:** 1Instituto de Física, Universidade Federal de Goiás, Goiânia 74690-900, GO, Brazil; norton@ufg.br; 2Departamento de Física, Universidade Federal de São Carlos, São Carlos 13565-905, SP, Brazil; mdciro@df.ufscar.br (C.M.D.); celsovb@df.ufscar.br (C.J.V.-B.)

**Keywords:** quantum thermodynamics, cavity quantum electrodynamics, Ramsey zone

## Abstract

We studied the thermodynamic properties such as the entropy, heat ($J_{Q}$), and work ($J_{W}$) rates involved when an atom passes through a Ramsey zone, which consists of a mode field inside a low-quality factor cavity that behaves classically, promoting rotations on the atomic state. Focusing on the atom, we show that $J_{W}$ predominates when the atomic rotations are successful, maintaining its maximum purity as computed by the von Neumann entropy. Conversely, $J_{Q}$ stands out when the atomic state ceases to be pure due to its entanglement with the cavity mode. With this, we interpret the quantum-to-classical transition in light of the heat and work rates. Besides, we show that, for the cavity mode to work as a Ramsey zone (classical field), several photons (of the order of $10^{6}$) need to cross the cavity, which explains its classical behavior, even when the inside average number of photons is of the order of unity.

## 1. Introduction

Cavity Quantum Electrodynamics (CQED) studies the interaction between light confined in cavities and atoms where the quantum nature of light and atoms is significant [1,2,3]. In CQED, quantum operations reach the individual control of atomic levels and their interaction with a single photon for engineering quantum states, in particular the qubits (two-level systems) that are now being applied in the construction of quantum computers [4].

The specific case of a qubit composed by a single two-level atom interacting with a cavity mode field is described, in the rotating-wave approximation (valid when the atom–field coupling is much weaker than their natural oscillation frequency) by the Jaynes–Cummings model [5], which promotes the so-called Rabi oscillations $|g\rangle |n\rangle ↔|e\rangle |n+1\rangle$ between the atom ground ($|g\rangle$) and excited ($|e\rangle$) states and the cavity state ($|m\rangle$) through the interaction term $H_{JC}=\hbar g\left(\sigma_{+}a+\sigma_{-}a^{†}\right)$, where $a^{†}$ (*a*) is the creation (annihilation) operator for the cavity mode field and $\sigma_{+}$ ($\sigma_{-}$) is the raising (lowering) operator for the atom, while *g* describes the strength of the Rabi frequency. If the cavity is on resonance with the atomic transition $|g\rangle ↔|e\rangle$, among the fundamental operations between the atom and the field, we can highlight, for example, the one in which a π/2 pulse promotes a coherent exchange of photons between the state of the atomic qubit and the cavity mode field qubit, resulting in the evolution $\left(\alpha |g\rangle +\beta |e\rangle \right)|0\rangle ↔|g\rangle \left(\alpha |0\rangle +\beta |1\rangle \right)$. This type of interaction leaves the field state inside the cavity in a superposition of vacuum and one-photon states. Likewise, π/4 pulses starting from $|e\rangle |0\rangle$ result in the maximum entangled state $\left(|e\rangle |0\rangle -i|g\rangle |1\rangle \right)/\sqrt{2}$. On the other hand, atomic superposition states as $\alpha |e\rangle +\beta |g\rangle$ can only be obtained using some classical resource, such that the initial cavity mode $|\psi_{i}\rangle$ and atomic $|a\rangle$ states emerge, at the end of the operation, in a product state $|a\rangle |\psi_{i}\rangle \to \left(\alpha |g\rangle +\beta |e\rangle \right)|\psi_{f}\rangle$.

In the microwave frequency domain, for instance, such a superposition of atomic states is generated using the so-called Ramsey zone [2,3,6,7], which is schematically illustrated in Figure 1a. This technique employs a low-quality factor cavity cooled to near absolute zero [2,3,8], which is continuously pumped by an external source modeled by $H_{p}=\varepsilon \left(e^{i\omega_{p}t}a+e^{-i\omega_{p}t}a^{†}\right)$, with $\omega_{p}$ and ε being, respectively, the frequency and the strength of the driving field, to compensate for the relatively short lifetimes of the photons, such that the cavity mode field is described by a coherent steady state [9,10]. Interestingly, even when the low-quality factor cavity has one photon on average, thus stressing the quantum character of the cavity mode field [11], the action of the external field, which is the classic resource necessary for the success of this interferometric technique, in addition to the strong cavity–field dissipation, produces an effective atom–field interaction that results in a pure atomic rotation, i.e., without entanglement with the cavity–field mode states [2], on the atomic state only, without the atom–field entanglement that one would expect from purely atomic quantum states or states of maximum purity, i.e., with null von Neumann entropy.

In this work, we studied the physics of the Ramsey zone [8,10,11] from the perspective of the burgeoning field of quantum thermodynamics, which has increasingly attracted the attention of researchers in recent decades [12,13,14,15,16,17,18,19,20,21,22,23,24]. To this aim, we focused on the atom as the system of interest to quantify the amount of heat $J_{Q}$ and work $J_{W}$ rates involved during the atom–field–reservoir interaction [25,26]. The purity of the state is quantified by von Neumann entropy, which vanishes for pure states and is maximum for maximally mixed states. Furthermore, the von Neumann entropy has the remarkable property of being maximized by Gibbs states, which describe systems in thermodynamic equilibrium. In the present study, where we deal with an out-of-equilibrium system, von Neumann entropy provides information about the purity of the atomic state, which is our system of interest. As we shall see, while the work is associated directly with the field, a certain amount of work is also indirectly associated with the pure atomic rotation, since heat alone, by its very definition, would not be capable of producing an operation that results in coherence in the final states. As discussed below, a key ingredient in all of this discussion is a unitary transformation that allows us to work on a displaced picture of the field in the cavity, thus enabling us to easily identify the work and the heat rates on the atom. Furthermore, when studying the thermodynamic features of the Ramsey zone, we will demonstrate that, although, on average, there is only one photon inside the cavity, for a pure atomic rotation to take place, it is absolutely necessary that millions of photons enter and exit the cavity during the time necessary to produce the desired rotation. This huge amount of photons corresponds to an energy that is absurdly greater than the work actually needed to produce just the atomic rotation.

In the next section (Section 2), we introduce the model that describes the Ramsey zone and present a method for calculating the work associated with the atom. In Section 3, we present our results concerning the heat and work rates [25] in our system and revisit the discussion of how the classical behavior of the field can occur in a cavity that contains on average only one photon. As we will show, for the system to behave classically, it is necessary that a large amount of photons cross the cavity. Finally, in Section 4, we present our conclusions.

## 2. Model

The dynamics of a Ramsey zone is described by a master equation composed of a unitary and a dissipative part [27]. The unitary part is governed by the Jaynes–Cummings and driving field Hamiltonians (ℏ=1) [10,11]:(1)$$H=\omega \left(a^{†}a+\frac{1}{2}\right)+\frac{\omega }{2}\sigma_{z}+\left(g\sigma_{+}a+\varepsilon e^{i\omega t}a+H.c.\right),$$
where the first term describes the cavity mode field of frequency ω, the second term describes the two-level atom on-resonance with the cavity mode field with $\sigma_{z}=|e\rangle \langle e|-|g\rangle \langle g|$, the third term describes the atom–cavity mode field interaction with $g=g^{*}$ being the Rabi frequency, and the fourth term describes the resonant pumping ($\omega_{p}=\omega$) on the cavity mode field. This fourth term, as it appears in the equation above, indicates that the work is associated with the cavity–field rather than directly with the atom, which is our thermodynamic system of interest, and H.c. stands for the Hermitian conjugate. To obtain a Hamiltonian that reveals the work that is indirectly associated with the atom, we proceed as follows. First, we move to a rotating frame according to the interaction picture, such that $H\to H_{I}$, with
(2)$$H_{I}=g\left(\sigma_{+}a+\sigma_{-}a^{†}\right)+\varepsilon \left(a+a^{†}\right).$$Assuming a weak interaction between the cavity mode field (atom) and the reservoir modes, a weak interaction between the cavity mode field and the atom (g≪ω), a weak driving (ε≪ω), and taking into account that experiments are performed by cooling the system to near absolute zero, the dynamics of the whole system is governed by the master equation in the interaction picture [27]:(3)$${\dot{\rho }}_{I}=-i\left[H_{I},\rho_{I}\right]+\kappa L_{a}\left(\rho_{I}\right)+\gamma L_{\sigma_{-}}\left(\rho_{I}\right),$$
where κ (γ) is the cavity mode field (atom) dissipation rate and $L_{\beta }\left(\rho_{I}\right)=2\beta \rho_{I}\beta^{†}-\beta^{†}\beta \rho_{I}-\rho_{I}\beta^{†}\beta$ ($\beta =a,\sigma_{-}$) [10,11]. After that, following [10], first, we write the master equation in the displaced picture by applying the time-independent unitary operation $D\left(\alpha \right)=\exp\left(\alpha a^{†}-\alpha^{*}a\right)$, with α=−iε/κ such that
(4)$${\dot{\tilde{\rho }}}_{I}=-i\left[H_{JC}+H_{SC},{\tilde{\rho }}_{I}\right]+\kappa L_{a}\left({\tilde{\rho }}_{I}\right)+\gamma L_{\sigma_{-}}\left({\tilde{\rho }}_{I}\right),$$
with ${\tilde{\rho }}_{I}=D^{†}\left(\alpha \right)\rho_{I}D\left(\alpha \right)$, $H_{JC}=g\left(\sigma_{+}a+\sigma_{-}a^{†}\right)$, and $H_{SC}=\alpha g\sigma_{+}+\alpha^{*}g\sigma_{-}$. The equation above allows us to identify an effective classical field driving the atomic state, thus capable of associating work with the atom [25], and another part, which can introduce non-unitarity to the evolution of the atom, which consists, in addition to the atomic decay, of interaction with the mode of the cavity together with the cavity dissipation. This can be clearly seen when we deal with effective dynamics, by tracing over the cavity mode variables. For instance, considering $g\ll \kappa \sqrt{\langle n_{c}\rangle +1}$, where $\langle n_{c}\rangle$ is the intra-cavity average number of photons (in the displaced picture), we can adiabatically eliminate the field operators to obtain the effective master equation to the atom only in the Schrödinger picture [10]:(5)$${\dot{\rho }}_{at}=-i\left[H_{at},\rho_{at}\right]+\Gamma_{eff}L_{\sigma_{-}}\left(\rho_{at}\right),$$
where $H_{at}=\left(\omega /2\right)\sigma_{z}-\left(i\varepsilon g/\kappa \right)\left(\sigma_{+}e^{-i\omega t}-\sigma_{-}e^{i\omega t}\right)$ and $\Gamma_{eff}=g^{2}/\kappa +\gamma$.

From the master equation displayed in Equation (Equation 5), we can directly calculate the atomic heat ($J_{Q}$) and work ($J_{W}$) rates using Alicki’s definitions (in the Schrödinger picture) [25]:(6)$$J_{Q}=\mathrm{tr}\left(H_{at}{\dot{\rho }}_{at}\right)$$
and
(7)$$J_{W}=\mathrm{tr}\left({\dot{H}}_{at}\rho_{at}\right).$$According to these definitions, the rate of change of the Hamiltonian is related to work, while the dissipative part of the master equation is associated with heat. This approach inspired the deduction of Equation (Equation 5), in which we were able to describe the dynamics of the atom through an effective master equation, separating the unitary part from the dissipative one. Thus, from Equation (Equation 5), we can directly calculate how much of the heat and work rates are needed to make the Ramsey zone work properly, i.e., without generating entanglement between the atom and the cavity–field mode states. With these definitions, $J_{Q(W)}>0$ ($J_{Q(W)}<0$) means the two-level atom is receiving (losing) energy. Although the definition of heat and work remains a topic of discussion (see [28,29,30]), Alicki’s definitions are the most widely used in the context of master equations, which is why we used them in the present study.

Note that, for γ=0 and *g* small enough, the dynamics is equivalent to that of an atom pumped by a classical field, being approximately unitary, and therefore, only work is associated with the atom during the evolution. On the other hand, by increasing *g*, the dissipation term can take part in the dynamics, thus allowing the atom to exchange heat with its environment. However, depending on the strength of the driving field ε, one can have either work or heat flux dominating the dynamics (see the discussion in the next section). Furthermore, it is worthwhile to mention that one of the central points in thermodynamics is the definition of the system. By defining, therefore, what a system is and what its surroundings are, we are able to calculate the energy flows into and out of the system both in the form of heat and work. In the model we are considering here, the atom is our system of interest, while the cavity–field, the laser field, and the thermal reservoirs are the surroundings forming the effective thermal bath and external force for the atom. Having this in mind, note that Equation (Equation 4) must be consistent with the effective master equation (Equation (Equation 5)), thus producing the same result either for the heat $J_{Q}$ or work $J_{W}$ rates [25]. Although useful for understanding the general aspects of our system, both the heat and work rates calculated using Equation (Equation 5) will be limited in scope because of the approximation we have made. However, if we now rewrite Equation (Equation 4) as
(8)$${\dot{\rho }}_{at}=-i\left[H_{at},\rho_{at}\right]+L_{eff}\left(\rho_{at}\right),$$
where $\rho_{at}=U_{0}{\mathrm{tr}}_{f}\left({\tilde{\rho }}_{I}\right)U_{0}^{†}$ and $L_{eff}\left(\rho_{at}\right)=U_{0}{\mathrm{tr}}_{f}\left\{-i\left[H_{JC},{\tilde{\rho }}_{I}\right]+\kappa L\left(a\right){\tilde{\rho }}_{I}+\gamma L\left(\sigma_{-}\right){\tilde{\rho }}_{I}\right\}U_{0}^{†}$, with $U_{0}=e^{-i\omega \sigma_{z}/2}$, then we can clearly identify the terms responsible for both the heat and work exchange between the atom and its surroundings. This is an important remark, since, while Equation (Equation 5) restricts us to the regime $g\ll \kappa \sqrt{\langle n_{c}\rangle +1}$, Equation (Equation 8) allows us to numerically investigate the heat $J_{Q}$ and work $J_{W}$ rates for all values of *g* and ε. To derive Equation (Equation 8), we first trace Equation (Equation 4) over the cavity mode field and, then, transition back to the Schrödinger picture. Here, the numerical calculation is performed using the quantum optics toolbox [31,32], which allows us to easily integrate the master equation of our system and calculate the desired quantities.

## 3. Results

### 3.1. Heat and Work Fluxes in a Ramsey Zone

To investigate how the heat and the work rates behave, let us first analyze some extreme cases, where our intuition can help us to understand our system. To this end, we assumed the initial state in the displaced picture as $|e\rangle |0\rangle$ (i.e., the atom initially prepared in the excited state and the cavity mode in the vacuum, which represents a coherent state that would be reached in the steady state in the case without the atom and in the Schrödinger picture).

The first case is the one where the atom–field coupling *g* is much smaller than the cavity decay rate, i.e., g≪κ, and for appreciable driving strength ε, the dynamics of the system of whic is governed by Equation (Equation 5). In this case, the Jaynes–Cummings dynamics can be ignored, and therefore, there will be no entanglement of the atom with the cavity–field. This is clearly seen in Figure 1b, since the atom performs a complete rotation from the excited to ground state (see the evolution of $\langle \sigma_{z}\rangle$), keeping the von Neumann entropy $S=-\mathrm{tr}[\rho_{at}\ln\left(\rho_{at}\right)]$ equal to 0 during the whole evolution. In this case, as we see in Figure 1d, the heat flux $J_{Q}$ is null, while the work flux is non-null. Thus, the work flux $J_{W}$ involved in the atomic rotation has been completely directed to successfully accomplish this task.

In the second case, for intermediate or strong atom–field coupling, i.e., g≳κ and $\left|\varepsilon /\kappa \right|\ll 1$, the Jaynes–Cummings dynamics dominates and the atom–cavity mode state becomes entangled, resulting in a mixed atomic state after tracing over the field variables. Thus, only heat $J_{Q}$ is exchanged between the atom and its surroundings. This behavior can be seen in Figure 1c, where the von Neumann entropy achieves its maximum value (S=ln(2)) during the atomic evolution and the work (heat) flux is null (non-null), as can be seen in Figure 1e.

Outside of the extreme cases above, i.e., for neither too small nor too large $\left|g/\kappa \right|$, and intermediate values of $\left|\varepsilon /\kappa \right|$, the atom can perform work on the cavity–field, as well as become entangled with it, thus indicating that both the heat $J_{Q}$ and work $J_{W}$ rates are being exchanged with its surroundings. To analyze the thermodynamics of those cases, we considered the evolution of the atomic state until it reaches a population in the excited state $P_{e}$ equal to the population in the ground state $P_{g}$, that is when $\langle \sigma_{z}\rangle =0$, considering the atom–field initially in the state $|e\rangle |0\rangle$ (in the displaced picture). Then, at that time, we calculate the heat and work rates for the atom and the von Neumann entropy *S* for the atomic state. We calculated those quantities as a function of g/κ for different values of driving strengths ε/κ, and the results are shown in Figure 2. As expected, for any non-null value of ε/κ, the von Neumann entropy starts at minimal values for g→0 since, in this case, the influence of the quantum nature of the cavity mode on the atom is negligible (thus resulting in pure rotations on the atomic state only, i.e., without entanglement with the cavity mode field), and it increases for g→∞. However, its maximum value depends on the driving strength ε/κ: for weak driving, the von Neumann entropy reaches the maximum value S=ln(2) for stronger atom–field couplings, as we see in Figure 2a, for ε/κ=0.25. This happens because, in this limit, we have basically an atom initially in the excited state interacting with a cavity mode in the vacuum, whose final state is a maximally entangled one: $\left(|e\rangle |0\rangle -i|g\rangle |1\rangle \right)/\sqrt{2}$. On the other hand, for higher values of the atom–field coupling and stronger driving strengths, the cavity mode reaches a coherent state (with a non-null amplitude), which does not lead to a maximally atom–field entangled state anymore, as we see in Figure 2c, ε/κ=2.

Focusing on the heat and work rates for the atom, we can see that, for a fixed ε, as we increase the atom–field coupling g/κ, the normalized heat ($J_{Q}/\hbar g\omega$) and work ($J_{W}/\hbar g\omega$) rates always increase and decrease in modulus, respectively. However, a different thermodynamic behavior appears depending on the value of the driving strength. For low (high) values of ε/κ, we have (do not have) a crossing of the heat and work rates, as we see in Figure 2a (Figure 2c) for ε/κ=0.25 (ε/κ=2). By solving our system numerically, we were able to find the threshold between the crossing versus no crossing regimes, which happens for ε/κ≈1.1, as we see in Figure 2b. Thus, for ε/κ below the threshold, we can always find a range of atom–field coupling where the work flux goes to zero and the heat flux is dominant, thus characterizing a purely quantum regime (since this allows for a high degree of atom–field entanglement). On the other hand, for values of ε/κ above the threshold, both the work and heat rates are non-null (with $\left|J_{Q}\right|<\left|J_{W}\right|$ for all values of g/κ), making clear that both the quantum and classical aspects of the cavity–field contribute to the atomic dynamics. As discussed above, for g<κ, the effective dynamics is governed by Equation (Equation 5), which allows us to derive the heat and work rates expressions
(9)JQ=−2ωg2κ〈σ+σ−〉−2εg3κ2Im〈σ+〉,
(10)JW=−2ωεgκRe〈σ+〉.

Considering the evolution till $\langle \sigma_{+}\sigma_{-}\rangle =P_{e}=P_{g}=1/2$ and keeping only the terms proportional to ω (since, in the regime we are in, ω≫g), we can find the condition for the crossing point *C* in Figure 2a: $J_{Q}=J_{W}$, which reads
(11)gε=2Re〈σ+〉.For all g≪κ, we could numerically verify that Re〈σ+〉≃0.22, and then, g≃0.44ε is the point where the heat flux equals the work flux. In Figure 2a, for ε=0.25κ, the crossing point *C*, determined numerically, is given by g≃0.11κ, in total agreement with Equation (Equation 11). However, for g≳κ, Equation (Equation 5) is no longer valid and, then, the crossing point can no longer be analytically derived. However, even numerically, it is possible to identify three different regions according to the atom–field coupling, which do not depend on the driving field strength (see Figure 2): (I) the region in which g→0, where the atom–field entanglement is negligible and, therefore, the heat flux is also negligible and there is a steady behavior for the normalized work flux $J_{W}/\hbar g\omega$ (as g→0); in this region, the value of the normalized work flux depends only on the driving field strength: the higher the ε, the higher the module of normalized work flux $\left|J_{W}/\hbar g\omega \right|$ is; (II) the region where $10^{-1}≲g/\kappa ≲10^{1}$, in which both the normalized heat and work rates vary as a function of *g* (transient region); and finally, (III) the region in which g≫κ and the $J_{Q}/\hbar g\omega$ and $J_{W}/\hbar g\omega$ reach steady values again (as a function of *g*); in this region, both the normalized heat and work rates can exist, but their values depend again on the strength of the driving field: for ε/κ≪1, the normalized work flux goes to zero while the normalized heat flux becomes non-null. On the other hand, for ε/κ≫1, the normalized heat flux goes to zero while the normalized work flux becomes non-null. We can understand the behavior of the $J_{W}/\hbar g\omega$ and $J_{Q}/\hbar g\omega$ rates in Figure 2a–c in terms of entanglement generation. To this end, note from Figure 2a that, when the von Neumann entropy reaches its maximum value, meaning that the atom mode state approaches its maximum degree of entanglement, $\left|J_{Q}/\hbar g\omega \right|$ increases to its maximum, while $\left|J_{W}/\hbar g\omega \right|$ decreases to its minimum, to the point that $\left|J_{Q}/\hbar g\omega \right|$ becomes greater than $\left|J_{W}/\hbar g\omega \right|$. Interestingly, increasing ε prevents the maximum correlation from forming, as indicated by the von Neumann entropy in Figure 2b,c. As the von Neumann entropy stabilizes at a value lesser than the maximum possible (S=ln(2)) and, consequently, the correlation stabilizes at a value lower than its maximum possible, the heat generation is also limited, to the point that $\left|J_{Q}/\hbar g\omega \right|$ no longer exceeds $\left|J_{W}/\hbar g\omega \right|$, as indicated by Figure 2b,c, in which there is no crossover. Note the dual role of ε: it is responsible for (indirectly) increasing the work performed by the system (atom) and, at the same time, limits the creation of atom–field entanglement, thereby limiting the amount of heat that flows from the system (atom).

As stated earlier, as our system of interest is only the atom, we did not conduct a study detailing the work associated with the field or eventually stored in the atom–field correlations [33,34,35,36]. An investigation to compare the work associated directly with the field with the work associated indirectly with the atom, as well as with the atom–field correlations could shed some light on the role of the cavity in the extraction process of work from the atom. This will demand more refined definitions of thermodynamic quantities to describe the exchange of heat and work involving interacting subsystems evolving under independent reservoirs. This is an interesting point and will be investigated in the future.

### 3.2. On the Classical Behavior of the Cavity–Field

In this subsection, we discuss an important issue raised in [11], which is the fact that one can have, on average, only one photon inside the cavity, and even so, the cavity mode field can be treated from a classical perspective, since the atom–field entanglement that would be expected between two quantum systems does not occur. As we see in Figure 2, this happens when the heat flux becomes negligible, i.e., when g≪ε. To address this point more carefully, first note that during the time of the unitary operation, the pumping term $H_{p}=\varepsilon \left(a^{†}+a\right)$ is responsible for taking the cavity state from $|0\rangle$ to $|\alpha (t)\rangle =\exp\left[-i\varepsilon t(a^{†}+a)\right]|0\rangle$, with α=−iεt in Equation (Equation 2). Therefore, the amount of photons that cross the cavity while the unitary operation is taking place is given by ${\bar{n}}_{flux}={\left|\varepsilon t\right|}^{2}$. On the other hand, the number of photons that actually remain inside the cavity is given by the difference between what is pumped into the cavity and what leaks through the walls of the low-quality cavity, i.e., ${\bar{n}}_{cav}={\left|\varepsilon /\kappa \right|}^{2}$, resulting in ${\bar{n}}_{flux}=\kappa^{2}t^{2}{\bar{n}}_{cav}$. Therefore, if we require ${\bar{n}}_{cav}∼1$, then ${\bar{n}}_{flux}∼k^{2}t^{2}$. For the Ramsey zone to work properly, the κ≫g limit must be met. For cavities with a low-quality factor [2,3], we can estimate not only how many photons actually go through the cavity during the time the atom crosses the cavity, but how much energy $E=\hbar \omega n_{flux}$ must be invested for the unit operation *U* (i.e, the Ramsey zone) to be performed successfully. Now, if we note that a typical atom rotation occurs for εgt/κ∼π then ${\bar{n}}_{flux}∼\kappa^{2}/g^{2}$, and we can, therefore, estimate, assuming $\kappa /g∼10^{3}$, the total number of photons crossing the low-Q cavity as ${\bar{n}}_{flux}∼10^{6}$ photons. This huge amount of photons helps to understand the emergence of the classical behavior even when there is only one photon on average inside the cavity. As noted in [11], there is nothing special about choosing ${\bar{n}}_{cav}={\left|\varepsilon /\kappa \right|}^{2}∼1$: the unitary operation can be accomplished even for ${\bar{n}}_{cav}={\left|\varepsilon /\kappa \right|}^{2}\gg 1$, provided the requirement κ≫g is met. However, this would require increasing the velocity of the atom crossing the cavity. This large expenditure of energy ${\bar{E}}_{flux}=10^{6}\hbar \omega$ is in contrast to the energy ${\bar{E}}_{cav}$ that is actually needed to produce work on the cavity–field, as shown in Figure 2. To better appreciate how the total average number of photons inside the cavity changes with the parameters involved, in Figure 3, we show, in a log scale on the left, the number of photons crossing the cavity as a function of g/κ (purple solid line) for the evolution from $\langle \sigma_{z}\rangle ≃1$ till $\langle \sigma_{z}\rangle ≃0$ (π/2-rotation), which would represent, in a perfectly unitary evolution, a superposition state $\left(|e\rangle -i|g\rangle \right)/\sqrt{2}$. The von Neumann entropy (black dashed–dotted line) is also shown on the left scale. Note that, to produce with high purity a rotation of π/2, i.e., with von Neumann entropy close to zero, a high number of photons crossing the cavity is necessary, on the order of $10^{6}$.

## 4. Conclusions

In this work, using the approach employed in Section 2 to obtain Equations (Equation 5) and (Equation 8), we studied the thermodynamic quantities’ entropy *S*, heat $J_{Q}$, and work $J_{W}$ rates exchanged by an atom and its environment in the functioning of a Ramsey zone, which is a device employed to rotate atomic qubits. It is constituted by a field that interacts with the atom in a lossy cavity, whose energy is kept constant due to a pumping field resonant with the atom and the cavity mode. We showed that, for the parameters for which the Ramsey zone works properly, i.e., without entanglement generation and, therefore, causing the atomic state to evolve with a high degree of purity (minimal entropy), $J_{W}$ is the amount that stands out. On the other hand, for parameters for which the Ramsey zone fails to function, the atom state becomes highly entangled with the cavity mode field state, and therefore, there is a drastic decrease in the purity of the atomic state, as shown by the von Neumann entropy, in which case the amount that stands out is $J_{Q}$. Yet, for certain parameters where the degree of entanglement between the atom state and the cavity mode field state is finite, but not maximum and, therefore, the von Neumann entropy is neither minimum nor maximum, both heat and work are present. In addition, we demonstrated the existence of a specific value for ε beyond which entanglement generation no longer occurs and, consequently, no more heat can be extracted from the system (atom). Furthermore, our study revealed that the average amount of photons coming from the classical pump and crossing the lossy cavity during the Ramsey zone operation is of the order of millions of photons, which explains the classical behavior of the cavity mode, even if, on average, the lossy cavity contains only one photon. The results presented here provided a way to understand the dynamics of the Ramsey zone through quantum thermodynamics concepts, being of interest to the quantum optical and thermodynamics community. Hence, the Ramsey zone has potential application in quantum thermal machines that use the concepts of heat and work as a figure of merit for the calculation of efficiency and performance.

## Figures and Tables

**Figure 1 entropy-25-01430-f001:**
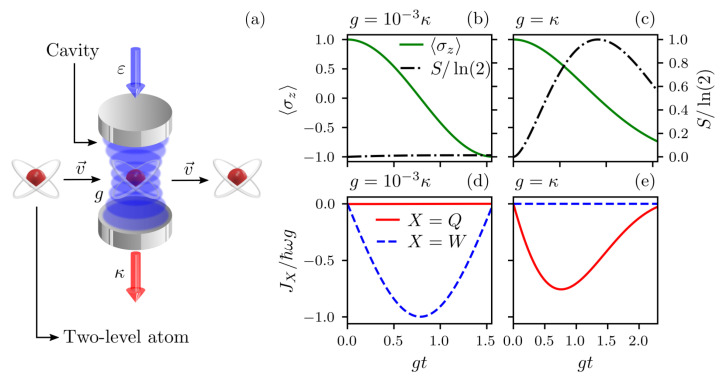
Thermodynamics of the Ramsey zone. (**a**) Experimental setup, where a two-level atom interacts (coupling *g*) with a single mode when passing (with velocity $\vec{v})$) through a low-quality factor cavity driven by an external source (strength ε) and with a decay rate κ. Panels (**b**,**c**) show the atomic population inversion $\langle \sigma_{z}\rangle$ and the normalized von Neumann entropy S/ln(2) as a function of gt. Panels (**d**,**e**) show the normalized work flux $J_{W}/\hbar \omega g$ and the normalized heat flux $J_{Q}/\hbar \omega g$ also as a function of gt. Panels (**b**,**d**) are for $g=10^{-3}\kappa$ and ε=κ, resulting in null heat flux and non-null work flux during the rotation process of the atomic state, thus characterizing a unitary evolution. On the other hand, Panels (**c**,**e**) are for g=κ and $\varepsilon =10^{-3}\kappa$, which results in non-null heat flux and null work flux during the rotation process of the atomic state, i.e., a purely non-unitary evolution (when tracing over the mode variables). In all plots, we neglected the atomic decay (γ=0) and considered |e〉|0〉 as the initial atom–field state (in the displaced picture).

**Figure 2 entropy-25-01430-f002:**
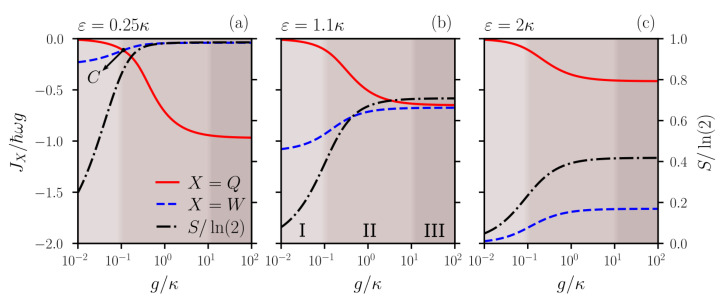
Normalized rates of heat $J_{Q}/\hbar \omega g$ (red solid line) and of work $J_{W}/\hbar \omega g$ (blue dashed line) on the atom and the normalized von Neumann entropy S/ln(2) (black dashed–dotted line) as a function of the atom–field coupling g/κ for different values of the driving field strength: (**a**) ε/κ=0.25, (**b**) ε/κ=1.1 (critical driving strength), and (**c**) ε/κ=2.0. The heat and work rates are computed during the evolution of the system from the initial state $|e\rangle |0\rangle$ (i.e., the atom initially prepared in the excited state and the cavity mode in the vacuum state, in the displaced picture, which represents a coherent state defined by the driving field in the laboratory frame), until the atom reaches null population inversion, i.e., $\langle \sigma_{z}\rangle =0$. For all values of ε/κ smaller (bigger) than the critical point ε/κ=1.1, the heat and work rates always cross (do not cross), indicating two different thermodynamic regimes. For g≪κ, the crossing point *C* is given by Equation (Equation 11). In these plots, we also identify three different regions: (I) the region g→0, where the atom–field entanglement is negligible and, therefore, the heat flux is also negligible; (II) the region where $10^{-1}≲g/\kappa ≲10^{1}$, in which both the normalized heat and work rates vary as a function of *g* (transient region); and (III) the region in which g≫κ and $J_{Q}/\hbar g\omega$ and $J_{W}/\hbar g\omega$ reach finite steady values (as a function of *g*), which depend on the driving strength ε.

**Figure 3 entropy-25-01430-f003:**
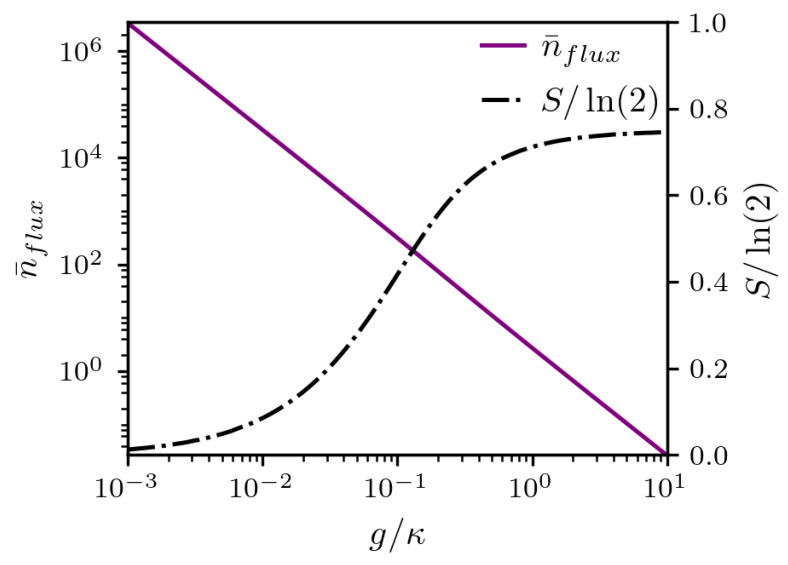
Normalized von Neumann entropy S/ln(2) (black dashed–dotted line) and average photon number ${\bar{n}}_{flux}$ (purple solid line) crossing the lossy cavity versus g/κ at time when the atom achieves the state for $P_{e}=\langle e|\rho_{at}|e\rangle ≃P_{g}=\langle g|\rho_{at}|g\rangle \approx 0.5$, i.e., the populations of the ground and excited states are the same for fixed ${\bar{n}}_{cav}={\left|\varepsilon /\kappa \right|}^{2}=1$, which is the average number actually found in the cavity.

## Data Availability

Not applicable.

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
