# Peer review of "Thermodynamics of the Ramsey Zone"

_entropy, 2023, doi:10.3390/e25101430_

Round 1

Reviewer 1 Report

In the paper, the authors study the thermodynamical properties of the system composed of an atom interacting with a driven low-quality factor cavity. They consider different regimes of the coupling $g$, the driving $\epsilon$, and the leakage $\kappa$. However, I think the paper lacks the derivation of the master equation, which generates confusion about the validity of the master equation.  Therefore, I don’t recommend it for publication. 

(1) If I don’t make mistakes, the master equation is the so-called local master equation, at most valid in the weak coupling case. If $g$ or $\epsilon$ are far larger than the leakage $\kappa$, the global master equation should be utilized.

(2) When the master equation is derived, the high-order terms of $\kappa$ have been omitted. In this case, $g$ far less than $\kappa$ makes no sense.

(3) If $\epsilon$ is relatively large, the master equation will become quite different. The authors have to consider the details of the derivation of the master equation.

(4) The authors consider the thermodynamical behaviors in a changed representation. To my knowledge, such a consideration could bring confusion. It should turn back to the original representation unless they are independent of representations.

(5) If the stronger coupling or driving is considered, the definitions of the thermodynamical quantities for local systems are confused.

Reviewer 2 Report

In this manuscript the authors study the dynamics of work and heat for the case of a two-level atomic system interacting with the radiation contained in a cavity QED, as proposed in a Ramsey interferometry scheme. Several regimes of atom-radiation coupling and dissipative effects are considered in the analysis. This study may add some interesting information to the ongoing study of quantum thermodynamics if the authors could address the following points:

1-    First, I don’t agree with the terminologies “work flux” and “heat flux” used by the authors. In physics, we use “flux of a quantity” when this quantity is crossing some area (surface). This is not the case here; the authors are actually talking about the rate in which work and heat change with time. Then, the correct terminology would be “work rate” and “heat rate”. In fact, this is how the authors of Ref. [Entropy 16, 3434 (2014)] address this same issue.

2-    How do the authors physically distinguish quantum work and heat? In general, the former is linked to a change in the energy levels of the system of interest and latter to a change in the energy level populations. About this issue, recent works (some of which also address the problem of the atom-radiation interaction) have been ignored. For example: [Phys. Rev. E 102, 062152 (2020)], [Sci. Rep. 13 160 (2023)], and [New J. Phys. 25 043019 (2023)].    

3-    How could the negative values of the entropy be explained in the graphics? The von Neumann entropy of a qubit must vary from 0 (pure state) to ln(2) (totally mixed state).

4-    A relatively long discussion after Eq. (9) is made on the points in which the so-called work flux and heat flux cross. Is there some physical importance related to these points? This is not specified.

5-    The authors start the conclusion by saying “In this work, using a method we developed, we studied...”. In my view, this article studies the atom-radiation dynamics from the master equation given in eq. (5), and use the solution to describe the thermodynamics by means of well-known results, say the equations for J_W and J_Q. Therefore, I don’t agree that the method used was developed by the authors.

6-    Given the importance of the equations for J_W and J_Q in the development of the work, I think they should be better emphasized.

Some corrections/typos:

“To the Ramsey zone work properly...” should be exchanged by “For the Ramsey zone to work properly...”

“...how many photons actually go trough the cavity...” should be exchanged by “...how many photons actually go through the cavity...”

Reviewer 3 Report

In this work, the authors present the thermodynamics of the Ramsey zone for a physical system modeled employing a Hamiltonian of the Jaynes-Cummings type.

This work shows quite interesting results since they can identify the classical or quantum regime with the evaluation of the work and heat of the system. If heat governs and work goes to zero (or is negligible), it is mentioned that the system is in a purely quantum regime. Both classical and quantum aspects contribute to the atomic dynamics when work and heat are found.

It is described with particular interest where there is a crossover between work and heat, and an analytical type solution is presented for \omega >> g to find this point. The von Neumann entropy is also offered where it is seen that for cases where \epsilon / \kappa < 1 reaches its maximum (ln (2)) when we are in a regime of g>> \kappa.

Here, my first question arises. Figures 2(b) and 2(c) show that the entropy drops considerably from its maximum value. As \epsilon /k grows, it becomes steeper and steeper. Is it possible to derive some condition in those regimes to know the asymptote to which the von Neumann entropy of the system will reach? 

The other question I have is about thermodynamic cycles. How would you apply this system to a quantum machine? Do you refer to quantum Otto cycle-type systems? What would be the analogous thermodynamic processes here?

What would be the analogous thermodynamic processes that you would propose here? 

The doubt arises because work and heat appear intertwined in the proposal, so I wonder if an Otto machine is the most appropriate.

Round 2

Reviewer 1 Report

The authors have addressed all my questions. I think the current version can be accepted for publication.

Reviewer 2 Report

The authors corrected/clarified all points previously raised in the paper. I can now recommend it for publication. 

Reviewer 3 Report

The authors have wholly answered my concerns and improved the writing.

The article in my opinion, can be published at this time.